# DUAL LoRA: ENHANCING LoRA WITH MAGNITUDE AND DIRECTION UPDATES

## ABSTRACT

Low-rank adaptation (LoRA) is one of the most popular methods among parameter-efficient fine-tuning (PEFT) methods to adapt pre-trained large language models (LLMs) to specific downstream tasks. However, the model trained based on LoRA often has an unsatisfactory performance due to its low-rank assumption. In this paper, we propose a novel method called Dual LoRA to improve the performance by incorporating an inductive bias into the original LoRA. Specifically, we separate low-rank matrices into two groups: the magnitude group to control whether or not and how far we should update a parameter and the direction group to decide whether this parameter should move forward or backward, to better simulate the parameter updating process of the full fine-tuning based on gradient-based optimization algorithms. We show that this can be simply achieved by adding a ReLU function to the magnitude group and a sign function to the direction group. We conduct several experiments over a wide range of NLP tasks, including natural language generation (NLG), understanding (NLU), and commonsense reasoning datasets on GPT-2, RoBERTa, DeBERTa, and LLaMA-1/2/3 as baseline models. The results show that we consistently outperform LoRA and its state-of-the-art variants with the same number of trainable parameters.

## 1 INTRODUCTION

Large language models (LLMs) have shown promising results on almost all natural language processing (NLP) tasks (Touvron et al., 2023a; Achiam et al., 2023) and other multi-modal tasks (Liu et al., 2024a), by adapting a well trained LLM to different downstream applications. Full fine-tuning (FFT) is a straightforward way to achieve this goal, but it requires tremendous computational resources and time to complete the fine-tuning process. Thus, parameter-efficient fine-tuning (PEFT) which updates a small fraction (less than 2%) of parameters has attracted more and more attention due to its low memory and time requirements.

Traditional PEFT methods include adapter tuning (Hu et al., 2023) which adds trainable tiny modules to adapt to downstream tasks, prompt tuning (Peng et al., 2024) that inserts learnable prompt vectors to the existing input, and low-rank adaptation (LoRA) (Hu et al., 2021a) which updates the original parameters by adding low-rank matrices. Among them, LoRA surpasses other methods by achieving better performance without generating additional inference costs.

Many follow-ups manage to improve the fine-tuning performance of LoRA. LoRA+ (Hayou et al., 2024) uses different learning rates to update low-rank matrices $A$ and $B$ and enhance the performance with a well-chosen learning rate ratio. DoRA (Liu et al., 2024b) decomposes the original weight matrix into a normalized matrix and its corresponding norm and applies the original LoRA to the normalized matrix. FLoRA (Si et al., 2024) generates LoRA to high dimensional space and inserts a low-rank core matrix into the original LoRA matrices to improve its performance. MoRA (Jiang et al., 2024) replaces the low-rank matrices with a square matrix to achieve high-rank updating and applies a compress layer and a decompress layer to maintain a roughly similar number of trainable parameters. However, they share a common drawback: as the trainable parameters are much fewer than those of FFT, updating them without incorporating prior knowledge will inevitably result in unsatisfactory model accuracy.

Thus, in this paper we introduce an inductive bias into the original LoRA method, *i.e.,* to simulate the parameter updating process of FFT, which utilizes gradient-based optimization algorithms.

Specifically, we divide the low-rank matrices into two groups: the magnitude group, which controls whether and to what extent a parameter should be updated; and the direction group, which determines the direction of the update——whether it should be positive or negative. The whole fine-tuning process can be treated as adjusting the sign and magnitude of each element in the update matrix and adding them back to the original parameters to gradually achieve the optimal solution. We conduct experiments to validate the effectiveness of our method over a wide range of NLP tasks including natural language generation (NLG), understanding (NLU), and commonsense reasoning to make a fair comparison with state-of-the-art methods. The evaluation results on different LLM models such as GPT-2, RoBERTa, DeBERTa, LLaMA-7B/13B, LLaMA2-7B, LLaMA3-8B, and LLaMA3-70B-Instruct show that we can achieve consistent improvements over these SOTA methods by using the same number of training parameters.

The contributions of our method are summarized as follows:

- We introduce Dual LoRA, a novel method that replaces the original low-rank matrices in LoRA with two groups of parameters: a magnitude group and a direction group to separately determine the amplitude and sign of the update to the original parameters in the LLMs. This can be treated as incorporating an inductive bias into the original LoRA to better learn the parameter updating process of FFT, which can improve the performance.

- Dual LoRA consistently outperforms state-of-the-art methods on a wide range of NLP tasks across various baseline models with different sizes (from 7B to up to 70B), which demonstrates the effectiveness of our method.

## 2 RELATED WORKS

In this section, we first introduce different parameter-efficient fine-tuning (PEFT) methods, followed by a deeper dive into the LoRA series methods.

### 2.1 PEFT METHODS IN LLMS

**Prefix tuning** is the first kind of methods (Li & Liang, 2021; Liu et al., 2022; Zhang et al., 2024) in PEFT. It was first proposed by Li *et.al.* (Li & Liang, 2021), which was a lightweight alternative to FFT that kept LLM parameters frozen and only optimized a sequence of continuous task-specific vectors called prefix. Dynamic prefix-tuning (Liu et al., 2022) proposed a generative template-based event extraction method with dynamic prefixes by integrating context information with type-specific prefixes to learn a context-specific prefix for each context. Selective prefix-tuning (Zhang et al., 2024) showed that prefix tokens carried context-specific information and enhanced their specialization can improve model performance. Thus, they integrated a selective mechanism inspired by selective self-attention and introduced selective loss to encourage diversity in prefix tokens.

**Prompt tuning** is the second kind of PEFT method that added trainable embeddings to original word embeddings and learned these soft prompts through back-propagation and tuned them to incorporate signals from any number of labeled examples (Lester et al., 2021). P-Tuning v2 (Liu et al., 2021) empirically found that properly optimized prompt tuning can be universally effective across a wide range of model scales and NLU tasks, which increased the capacity of continuous prompts and closed the gap to FFT. Knowledgeable Prompt-tuning (Hu et al., 2021b) improved and stabilized the original prompt-tuning method by expanding the label word space of the verbalizer with external knowledge bases and refining it with PLM before predicting.

**Representation fine-tuning (REFT)** aims to train interventions that manipulate model representations to steer model behaviors on downstream tasks at inference time. ReFT (Wu et al., 2024) introduced a family of ReFT methods that operated on a frozen base model and learned task-specific interventions on hidden representations.

Although the aforementioned methods improved the performance of LLMs in downstream tasks, they suffered the problem that the original architecture of the baseline model needed to be changed and the inference speed was slowed down. Compared to them, LoRA-based methods had exactly the same inference latency to the baseline LLMs.

## 2.2 LoRA-Based Methods

LoRA (Hu et al., 2021a) assumed that only a small number of task-specific parameters needed to be tuned to fit the downstream tasks and updated the weights with two low-rank matrices. These matrices can be merged back into the original weights during inference to avoid additional computational costs. LoRA+ (Hayou et al., 2024) argued that LoRA led to sub-optimal results, and the problem can be corrected by setting different learning rates for the low-rank matrices $A$ and $B$ with a fixed learning rate ratio. MoRA (Jiang et al., 2024) believed that the low-rank updating mechanism limited the ability of LLMs and used a square matrix to achieve high-rank updating with the same number of trainable parameters. Two non-parameter operators were used to reduce the input dimension and increase the output dimension of this square matrix. DoRA (Liu et al., 2024b) decomposed the pre-trained weight into magnitude and direction for fine-tuning, and employed original LoRA for direction component update to accelerate the training process.

The methods mentioned above can improve the performance of downstream tasks. However, their performance is still unsatisfactory because of the low-rank assumption (Hu et al., 2021a; Hayou et al., 2024; Liu et al., 2024b). Although MoRA (Jiang et al., 2024) attempted to address this issue by using a high-rank matrix, its rank and the number of trainable parameters remained significantly lower than those in FFT. Thus, it is difficult to achieve satisfactory model performance without incorporating prior knowledge into the training process.

Note that both DoRA and our method have magnitude and direction groups, but the meaning behind them is totally different. The direction and magnitude in DoRA can be treated as a normalized weight matrix and its corresponding norm. In our method, we are trying to simulate the parameter updating process of FFT which utilizes gradient-based optimization algorithms. Thus, the direction and magnitude control the sign and to what extent a parameter should be updated.

Another family of methods aim to modify the gradient calculation and backward propagation process of training, such as GaLore (Zhao et al., 2024), FLoRA (Hao et al., 2024) and GaRare (Liu et al.). These methods are orthogonal to the proposed Dual LoRA which only focuses on the architecture and forward pass modification, and a detailed discussion falls outside the scope of this paper.

## 3 Method

In this section, we first introduce the preliminaries of LoRA and optimization methods. Then, we give a thorough analysis of our proposed Dual LoRA and explain its advantage over previous LoRA-based methods.

### 3.1 Low-Rank Adaptation (LoRA)

Given a pre-trained weight matrix $W_0 \in \mathbb{R}^{d \times k}$, LoRA (Hu et al., 2021a) assumes that a low "intrinsic rank" is enough during adaptation on downstream tasks and constrains the updated matrix with a low-rank decomposition:

$$W' = W_0 + \Delta W = W_0 + \frac{\alpha}{r} \cdot BA, \tag{1}$$

where $B \in \mathbb{R}^{d \times r}$ and $A \in \mathbb{R}^{r \times k}$ are two low-rank matrices with rank $r \ll min(d, k)$, $\alpha$ is a fixed hyper-parameter to control the influence of the low-rank matrices, and $W'$ is the final weight matrix after fine-tuning.

Given an original forward pass $h = W_0 x$ with an input $x$, the modified forward pass can be expressed as:

$$h = W_0 x + \Delta W x = (W_0 + \frac{\alpha}{r} \cdot BA)x. \tag{2}$$

Note that the usage of LoRA does not affect the inference speed since the low-rank matrices $A$ and $B$ can be merged back into the original weight $W_0$, and the dimension of the final weight matrix $W'$ is the same as the pre-trained weight matrix $W_0$. Since the trainable low-rank matrices have fewer parameters (less than 2%) compared to the original matrices, LoRA usually has insufficient performance.

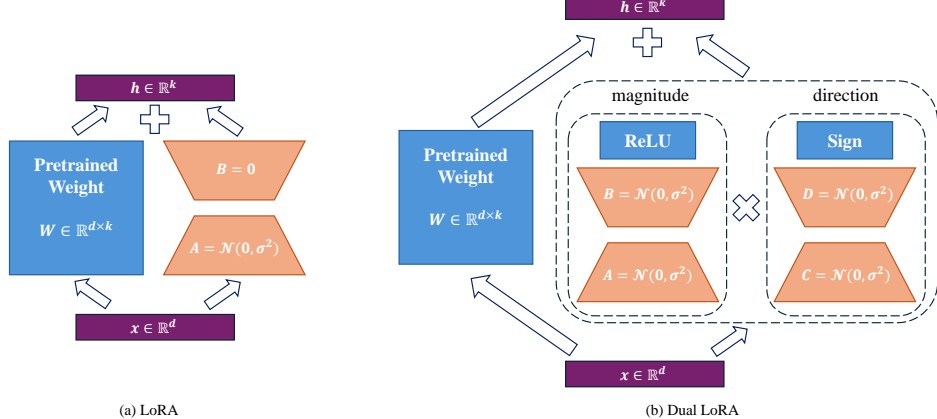

Figure 1: The architecture of the original LoRA and our proposed Dual LoRA. The low-rank update matrices are separated into the magnitude group and the direction group.

## 3.2 OPTIMIZATION METHODS

Given a loss function $\ell(\hat{y}, y)$ which measures the cost between the prediction $\hat{y}$ and the ground-truth label $y$, we can choose a family $\mathcal{F}$ of functions $f_w(x)$ with learnable weight $w$ and input $x$, and seek the function $f \in \mathcal{F}$ to minimize the loss $\ell(f_w(x), y)$ averaged on the input examples:

$$E_n(f_w) = \frac{1}{n} \sum_{i=1}^{n} \ell(f_w(x_i), y_i). \tag{3}$$

In order to minimize the empirical risk $E_n(f_w)$, a global optimum weight $w^*$ needs to be found step by step using a series of optimization methods. Specifically, we have:

$$w_{t+1} = w_t + \gamma \Delta w, \tag{4}$$

where $\gamma$ is the learning rate and $w_{t+1}$ is expected to converge to the global optimum $w^*$ as the training proceed.

To achieve this, different optimization methods leverage different ways to compute $\Delta w$. For example, gradient descent (Bottou, 2010) uses $\Delta w = \frac{1}{n} \sum_{i=1}^{n} \nabla_w \ell(f(x_i), y_i)$ to compute the update, and Adam (Kingma, 2014) uses $\Delta w = \hat{m}_t / (\sqrt{\hat{v}_t} + \epsilon)$ where $\hat{m}_t$ and $\hat{v}_t$ are the first-moment estimate and second-moment estimate, and $\epsilon = 10^{-8}$.

Both FFT and LoRA fine-tune the model based on the optimization methods mentioned above. However, FFT assumes $\Delta w$ is a full-rank matrix while LoRA decomposes $\Delta w$ into two low-rank matrices and trains them without any other prior knowledge, which is the main reason that causes the performance drop.

## 3.3 DUAL LoRA

Note that the update matrix $\Delta w$ can always be decomposed into magnitude and direction regardless of the optimization method used. Learning these components separately can be treated as adding an inductive bias into the original LoRA, aiding in facilitating the search for the optimal solution within the solution space.

Instead of using two low-rank matrices, we use four low-rank matrices and separate them into a magnitude group and a direction group in Dual LoRA, as shown in Fig. 1.

**Magnitude group.** Given two low-rank matrices $A \in \mathbb{R}^{r_1 \times k}$ and $B \in \mathbb{R}^{d \times r_1}$, the magnitude group can be computed as:

$$W_m = \text{ReLU}(BA), \tag{5}$$

which has two effects. Firstly, non-negative outputs can be treated as learning the magnitude of the update during the training process. Secondly, we can easily freeze some of the elements that are already well-trained for the downstream tasks in the original weight matrix by learning the output elements of $BA$ to be negative and filter them out with ReLU function, which is hard for previous LoRA-based methods to achieve such a goal.

**Direction group.** Given two low-rank matrices $C \in \mathbb{R}^{r_2 \times k}$ and $D \in \mathbb{R}^{d \times r_2}$, the direction group can be computed as:

$$W_d = \text{Sign}(DC), \tag{6}$$

where $\text{Sign}(\cdot)$ is an element-wise operation that outputs $+1$ for positive input and $-1$ otherwise. Note that the gradient of the sign function is zero almost everywhere, and backward propagation cannot be applied during training. Thus, given $x_b = Sign(x)$, the straight-through estimator (STE) method (Bengio et al., 2013) is introduced to compute its gradient as:

$$\frac{\partial \mathcal{L}}{\partial x} = \text{Clip}(\frac{\partial \mathcal{L}}{\partial x_b}, -1, 1), \tag{7}$$

in which $\mathcal{L}$ is the corresponding loss function for a downstream task and:

$$\text{Clip}(x, -1, 1) = \begin{cases} -1, & \text{if } x < -1, \\ x, & \text{if } -1 \leq x < 1, \\ 1, & \text{otherwise.} \end{cases} \tag{8}$$

The direction group can control the sign of each element in the update matrix, which is a two-way direction to decide whether the element in the original weight matrix should move forward or backward.

**Overall update.** Given a pre-trained weight matrix $W_0$, the overall update of our Dual LoRA can be expressed as:

$$W' = W_0 + \Delta W = W_0 + \frac{\alpha}{\sqrt{r_1 r_2}} W_m \odot W_d, \tag{9}$$

where $\odot$ represents an element-wise product (Hadamard product) between two matrices. Similarly, given the original forward pass $h = W_0 x$, the modified forward pass is:

$$h = W_0 x + \Delta W x = (W_0 + \frac{\alpha}{\sqrt{r_1 r_2}} W_m \odot W_d)x, \tag{10}$$

which does not affect the inference process as long as we merge $\Delta W$ into $W_0$.

**Initialization.** LoRA uses random Gaussian initialization for $A$ and zero for $B$ to make sure the update matrix is zero at the beginning of training, as shown in Fig. 1(a). In Dual LoRA, however, none of the low-rank matrices in the magnitude group should be initialized with zero. Otherwise, either all trainable parameters are dead, or we cannot achieve the goal that the update matrix is zero due to the $\text{ReLU}(\cdot)$ function and $\text{Sign}(\cdot)$ function.

Specifically, given

$$\Delta W = \frac{\alpha}{\sqrt{r_1 r_2}} \text{ReLU}(BA) \odot \text{Sign}(DC), \tag{11}$$

we can compute the gradient of the loss function $\mathcal{L}$ with respect to four low-rank matrices as:

$$\frac{\partial \mathcal{L}}{\partial A} = \frac{\partial \mathcal{L}}{\partial \Delta W} \cdot \frac{\alpha}{\sqrt{r_1 r_2}} B^\top \cdot \text{Sign}(DC) \cdot \mathbb{1}_{BA>0}, \qquad \frac{\partial \mathcal{L}}{\partial B} = \frac{\partial \mathcal{L}}{\partial \Delta W} \cdot \frac{\alpha}{\sqrt{r_1 r_2}} \text{Sign}(DC) \cdot \mathbb{1}_{BA>0} \cdot A^\top,$$

$$\frac{\partial \mathcal{L}}{\partial C} = \text{Clip}(\frac{\partial \mathcal{L}}{\partial \Delta W}, -1, 1) \cdot \frac{\alpha}{\sqrt{r_1 r_2}} D^\top \cdot \text{ReLU}(BA), \quad \frac{\partial \mathcal{L}}{\partial D} = \text{Clip}(\frac{\partial \mathcal{L}}{\partial \Delta W}, -1, 1) \cdot \frac{\alpha}{\sqrt{r_1 r_2}} \text{ReLU}(BA) \cdot C^\top, \tag{12}$$

where $\mathbb{1}$ is the indicator function.

It is easy to know that when setting $A = 0$ or $B = 0$, we will have $\mathbb{1}_{BA>0} = 0$ and $\text{ReLU}(BA) = 0$ and all four gradients in Eq. 12 are zeros which will cause the training process to be dead. Setting $C = 0$ or $D = 0$ will not result in such a problem, but it cannot achieve the goal that the update matrix Eq. 11 is zero at the beginning of training since $\text{Sign}(x)$ always outputs $+1$ or $-1$ depending on the input. Thus, during the experiments, we use random Gaussian initialization for all four low-rank matrices and apply a warm-up strategy for the first few training steps to make sure that $\Delta W = 0$ at the start.

Table 1: The results of the proposed Dual LoRA and other competitors with LLaMA-7B/13B, LLaMA2-7B, LLaMA3-8B and LLaMA3-70B-Instruct on commonsense reasoning datasets. For all matrices, higher is better.

| Model | Methods | Trainable Param. (%) | Commonsense Reasoning Datasets | | | | | | | | |
|---|---|---|---|---|---|---|---|---|---|---|---|
| | | | BoolQ | PIQA | SIQA | HellaS | WinoG | ARC-e | ARC-c | OBQA | Avg. |
| L-7B | Adapter-P | 3.54 | 67.9 | 76.4 | 78.8 | 69.8 | 78.9 | 73.7 | 57.3 | 75.2 | 72.2 |
| | LoRA ($r = 64$) | 1.64 | 67.3 | 79.0 | 76.3 | 76.6 | 78.8 | 74.5 | 59.3 | 77.4 | 73.6 |
| | DoRA ($r = 32$) | 0.84 | 68.7 | **83.3** | 79.4 | 85.5 | 81.3 | 80.8 | **66.0** | 78.8 | 78.0 |
| | DoRA ($r = 64$) | 1.65 | 68.9 | 82.1 | 77.4 | 75.9 | 80.0 | 80.0 | 64.8 | 81.0 | 76.3 |
| | Dual LoRA ($r = 32$) | 1.64 | **70.0** | 83.2 | **79.5** | **87.3** | **83.0** | **81.6** | 65.2 | **81.0** | **78.9** |
| L-13B | LoRA ($r = 32$) | 0.67 | 71.6 | 83.4 | 80.0 | 89.9 | 84.2 | 81.2 | 67.7 | 80.8 | 79.9 |
| | DoRA ($r = 16$) | 0.35 | 71.7 | 84.2 | 80.6 | 90.5 | **85.2** | 83.1 | 68.4 | 80.4 | 80.5 |
| | DoRA ($r = 32$) | 0.68 | 72.4 | **84.9** | **81.2** | 91.5 | 83.7 | 84.6 | 68.9 | 81.6 | 81.1 |
| | Dual LoRA ($r = 16$) | 0.67 | **72.5** | 84.2 | 79.9 | **92.7** | 83.8 | **84.8** | **72.4** | **83.2** | **81.7** |
| L2-7B | LoRA ($r = 16$) | 0.41 | 70.4 | 82.9 | 79.0 | 81.3 | 81.5 | 82.4 | 69.2 | 80.4 | 78.4 |
| | LoRA ($r = 32$) | 0.83 | 68.9 | 82.2 | 78.1 | 86.9 | 81.2 | 79.3 | 65.4 | 78.4 | 77.6 |
| | DoRA ($r = 16$) | 0.43 | 63.5 | 82.8 | 79.5 | **90.6** | 82.4 | 83.9 | 69.9 | 81.8 | 79.3 |
| | DoRA ($r = 32$) | 0.84 | 72.2 | **83.5** | **80.3** | 89.0 | 82.5 | 84.1 | 69.5 | 80.4 | 80.2 |
| | Dual LoRA ($r = 16$) | 0.83 | **72.3** | 83.3 | 79.8 | 89.8 | **84.6** | **84.8** | 70.2 | **82.8** | **81.0** |
| L3-8B | RandLoRA | 0.70 | **76.3** | 88.1 | 80.3 | 95.7 | 86.1 | 90.4 | 80.9 | **87.0** | 85.6 |
| | LoRA ($r = 16$) | 0.35 | 71.7 | 86.8 | 79.5 | 93.9 | 84.4 | 87.4 | 76.3 | 84.2 | 83.0 |
| | LoRA ($r = 32$) | 0.70 | 71.2 | 85.1 | 79.3 | 92.1 | 82.6 | 85.2 | 70.1 | 81.4 | 80.9 |
| | DoRA ($r = 16$) | 0.35 | 75.1 | 87.8 | 80.8 | 95.6 | 86.3 | 90.4 | 80.0 | 85.6 | 85.2 |
| | DoRA ($r = 32$) | 0.71 | 71.7 | 88.0 | 80.2 | 95.5 | **86.6** | **90.7** | 78.4 | 85.0 | 84.5 |
| | Dual LoRA ($r = 16$) | 0.70 | 75.5 | **89.2** | **81.4** | **95.8** | 86.0 | 90.5 | **81.1** | 86.6 | **85.8** |
| L3-70B | LoRA ($r = 16$) | 0.197 | 78.6 | 92.8 | 83.4 | 92.7 | 92.6 | 97.5 | 91.7 | 94.4 | 90.5 |
| | DoRA ($r = 16$) | 0.202 | 78.4 | 93.0 | 83.8 | 96.5 | 92.3 | **97.6** | **92.3** | 94.6 | 91.1 |
| | Dual LoRA ($r = 8$) | 0.197 | **81.4** | **94.0** | **84.4** | **97.9** | **93.6** | 97.3 | 91.0 | **95.2** | **91.9** |

## 4 EXPERIMENTS

In this section, we evaluate the effectiveness of the proposed Dual LoRA on various NLP tasks. We compare our methods with other PEFT competitors by fine-tuning LlaMA-7B/13B, LLaMA2-7B, LLaMA3-8B, and LLaMA3-70B-Instruct models on a series of commonsense reasoning datasets. Then, we explore the ability of our method on the neural language understanding (NLU) dataset GLUE by fine-tuning RoBERTa base/large and DeBERTa XXL. Furthermore, we conduct experiments on neural language generation (NLG) datasets including E2E NLG Challenge, DART and WebNLG using GPT2_M and GPT2_L as backbones (see Appendix A). All experiments above show that Dual LoRA can surpass other LoRA-based methods with the same or fewer trainable parameters and achieve state-of-the-art results. Finally, we analyze our method further by performing a series of ablation studies. In the following experiments, we set the rank of the magnitude group and the direction group as the same, *i.e.*, $r_1 = r_2 = r$ unless specified.

**Competitors.** We compare Dual LoRA with a series of baseline methods including LoRA-based methods (LoRA (Hu et al., 2021a), LoRA+ (Hayou et al., 2024), GaLore (Zhao et al., 2024), GaRare (Liu et al.), Delta-LoRA (Zi et al., 2023), CorDA (Yang et al., 2024), VeRA (Kopiczko et al., 2024), RandLoRA (Albert et al., 2025), and DoRA (Liu et al., 2024b)) and other PEFT methods (efficient adapter design with LayerNorm (Adapter-L) (Lin et al., 2020), parallel adapter tuning (Adapter-P) (He et al., 2021) and prefix-layer tuning (Prefix) (Li & Liang, 2021)).

### 4.1 COMMONSENSE REASONING

**Datasets and baseline models.** We evaluate Dual LoRA and different PEFT methods on the commonsense reasoning task which is composed of eight different sub-tasks including BoolQ (Clark et al., 2019), PIQA (Bisk et al., 2020), Social IQa (Sap et al., 2019), HellaSwag (Zellers et al.,

Table 2: The results of the proposed Dual LoRA and other competitors with RoBERTa base/large and DeBERTa XXL on GLUE datasets. For all matrices, higher is better.

| Model | Methods | Trainable Param. (M) | GLUE | | | | | | | | |
|---|---|---|---|---|---|---|---|---|---|---|---|
| | | | MNLI | SST-2 | MRPC | CoLA | QNLI | QQP | RTE | STS-B | Avg. |
| RoB$_{base}$ | FFT | 125.0 | 87.6 | 94.8 | 90.2 | 63.6 | 92.8 | **91.9** | 78.7 | 91.2 | 86.4 |
| | GaLore ($r = 8$) | 0.3 | 87.2 | 94.4 | 92.0 | 61.8 | 92.3 | 91.2 | 79.1 | 90.8 | 85.9 |
| | GaRare ($r = 8$) | 0.3 | 87.2 | 94.4 | 91.5 | 61.1 | 92.3 | 90.9 | 79.3 | 90.3 | 85.9 |
| | Delta-LoRA ($r = 8$) | 0.3 | 87.5 | 95.1 | 90.2 | 63.8 | 93.1 | 90.9 | 87.0 | 91.6 | 87.4 |
| | CorDA ($r = 128$) | 21 | - | 93.1 | 89.7 | 59.6 | 91.5 | - | **88.1** | 90.2 | - |
| | VeRA | 0.3 | - | 91.9 | 88.4 | 59.9 | 90.5 | - | 74.9 | 90.4 | - |
| | RandLoRA | 0.7 | - | 92.2 | 88.0 | 59.4 | 91.3 | - | 74.7 | 90.3 | - |
| | LoRA ($r = 8$) | 0.3 | 87.0 | 94.6 | 89.2 | 60.9 | 92.9 | 90.7 | 92.0 | 91.1 | 86.1 |
| | LoRA ($r = 16$) | 0.6 | 87.0 | 95.1 | 89.0 | 63.9 | 93.0 | 91.2 | 83.4 | 91.1 | 86.7 |
| | LoRA+ ($r = 16$) | 0.6 | **87.8** | 95.2 | 90.4 | 65.9 | 92.6 | 91.2 | 82.3 | 91.4 | 87.1 |
| | DoRA ($r = 16$) | 0.6 | 87.7 | 95.3 | 87.8 | 64.8 | 92.6 | 90.8 | 82.2 | 90.8 | 86.5 |
| | Dual LoRA ($r = 8$) | 0.6 | **87.8** | **95.8** | **91.7** | **67.8** | **93.3** | 90.7 | **88.1** | 91.7 | **88.3** |
| RoB$_{large}$ | FFT | 355.0 | 90.2 | **96.4** | 90.9 | 68.0 | 94.7 | **92.2** | 86.6 | 92.4 | 88.9 |
| | GaLore ($r = 16$) | 1.6 | 90.8 | 96.1 | 91.7 | 68.3 | 95.7 | 91.9 | 87.0 | 92.5 | 89.3 |
| | GaRare ($r = 16$) | 1.6 | **91.3** | 96.2 | 91.7 | 67.9 | 94.6 | 91.8 | 87.4 | 92.3 | 89.2 |
| | VeRA | 0.3 | - | 95.8 | 89.3 | 65.3 | 94.1 | - | 81.6 | 91.8 | - |
| | RandLoRA | 1.8 | - | 95.5 | 90.1 | 67.4 | 94.1 | - | 84.5 | 91.4 | - |
| | LoRA ($r = 8$) | 0.8 | 90.2 | 96.5 | 89.5 | 63.8 | 94.5 | 91.5 | 88.8 | 92.5 | 88.3 |
| | LoRA ($r = 16$) | 1.6 | 90.2 | 95.9 | 90.9 | 66.0 | 94.4 | 91.6 | 87.4 | 92.3 | 88.6 |
| | LoRA+ ($r = 16$) | 1.6 | 90.3 | 96.3 | 91.4 | 68.7 | 94.7 | 91.6 | 88.8 | 92.5 | 89.3 |
| | DoRA ($r = 16$) | 1.6 | 90.5 | 96.2 | 89.7 | 68.5 | 92.6 | 91.5 | 89.2 | 92.3 | 88.8 |
| | Dual LoRA ($r = 8$) | 1.6 | 90.5 | **96.4** | **91.9** | **70.2** | **95.1** | 91.2 | **89.5** | **92.6** | **89.7** |
| DeB$_{XXL}$ | FFT | 1500.0 | 91.8 | 97.2 | **92.0** | 72.0 | 96.0 | **92.7** | 93.9 | 92.9 | 91.1 |
| | LoRA ($r = 16$) | 4.7 | 91.7 | 96.6 | 89.7 | 70.8 | 95.7 | 92.6 | 95.0 | 92.4 | 90.6 |
| | LoRA ($r = 32$) | 9.4 | **92.0** | **97.5** | 91.2 | 68.7 | 96.0 | 91.9 | 92.8 | 92.4 | 90.3 |
| | LoRA+ ($r = 32$) | 9.4 | 91.7 | **97.5** | 91.2 | 68.7 | 96.0 | 92.3 | 94.6 | 92.4 | 90.5 |
| | DoRA ($r = 32$) | 9.4 | 91.9 | 96.9 | 90.9 | 71.2 | 95.8 | 92.3 | 92.6 | 92.3 | 90.5 |
| | Dual LoRA ($r = 16$) | 9.4 | 91.9 | 97.1 | 91.9 | **74.0** | **96.2** | 92.6 | **95.3** | **93.4** | **91.6** |

2019), WinoGrande (Sakaguchi et al., 2021), ARC-easy/challange (Clark et al., 2018) and Open-BookQA (Mihaylov et al., 2018). Similarly to DoRA, we merge the training sets from all sub-tasks to get the final training set and perform evaluations on their own testing datasets for each task.

For the baseline models, we use LLaMA-7B/13B (Touvron et al., 2023a), LLaMA2-7B (Touvron et al., 2023b), LLaMA3-8B (Dubey et al., 2024), and LLaMA3-70B-Instruct (Dubey et al., 2024). We halve the rank of our low-rank matrices to ensure that the same number of trainable parameters are used compared to other LoRA-based methods. We tune the learning rate for our method, and all other training hyper-parameters are kept unchanged as in DoRA in order to make a fair comparison. We train one epoch for LLaMA3-70B-instruct, and three epochs for other baseline models.

**Results.** The results in Tab. 1 show that we can consistently outperform DoRA with less trainable parameters on all of the baseline models. For example, Dual LoRA enhances the average accuracy by 0.9%/0.6% compared to the previous best result on LLaMA-7B/13B. The performance gains are still notable on LLaMA2-7B, LLaMA3-8B and LLaMA3-70B-Instruct, which are 0.8%, 0.2% and 0.8%.

## 4.2 NEURAL LANGUAGE UNDERSTANDING (NLU)

**Datasets and baseline models.** We evaluate our method on a widely used natural language understanding dataset GLUE. It consists of eight different datasets includes MNLI (Williams et al., 2017), SST-2 (Socher et al., 2013), MRPC (Dolan & Brockett, 2005), CoLA (Warstadt, 2019), QNLI (Rajpurkar et al., 2018), QQP, RTE, and STS-B (Cer et al., 2017). The diversity makes the GELU benchmark a robust dataset for evaluating LLMs on NLU tasks.

For the baseline models, we use RoBERTa base/large (Liu, 2019) and DeBERTa XXL (He et al., 2020) as pretrained baseline models from the HuggingFace Transformers library (Wolf et al., 2020). Similarly to LoRA, we initialize the model to the LoRA-adapted MNLI checkpoint for MRPC, RTE, and STSB rather than the pre-trained baseline model. All other training parameters are the same as LoRA except for the learning rate.

**Results.** As shown in Tab. 2, the proposed Dual LoRA shows state-of-the-art results on all three baseline models. For example, on the small model RoBERTa base we can defeat previous methods LoRA, LoRA+, and DoRA by 1.6%, 1.2%, and 1.8% average accuracy. Similarly, on medium-sized model RoBERTa large, our Dual LoRA surpasses LoRA, LoRA+, and DoRA by 1.1%, 0.4%, and 0.9%. On DeBERTa XXL model with over 1500M total parameters, Dual LoRA can still exceed LoRA, LoRA+, and DoRA by 1.3%, 1.1%, and 1.1%. Note that we can even surpass the FFT methods on these baseline models by 1.9%, 0.8%, and 0.5%, which shows the priority of the proposed method.

## 4.3 ABLATION STUDY

We conduct several ablation studies to further verify the effectiveness of the proposed method.

**Dealing with the sign function.** In the previous section and experiments, we use the straight-through estimator (STE) method (Bengio et al., 2013) to compute the gradient of the sign function. Note that the sign function is a standard operator in the area of binary neural networks (BNNs), and there are many studies on dealing with the forward and backward passes of the sign function. For example, XNOR-Net (Rastegari et al., 2016) scales the weights after binarized:

$$\textbf{Forward:}\ x_b = \text{Sign}(x) \times \mathbf{E}_F(|x|), \quad \textbf{Backward:}\ \frac{\partial \mathcal{L}}{\partial x} = \frac{\partial \mathcal{L}}{\partial x_b}, \quad (13)$$

where $\mathbf{E}_F(|x|)$ is the mean of the absolute value of each output channel of weights. Dorefa-Net (Zhou et al., 2016) uses a constant scalar to scale all of the weights instead of doing channel-wise scaling:

$$\textbf{Forward:}\ x_b = \text{Sign}(x) \times \mathbf{E}(|x|), \quad \textbf{Backward:}\ \frac{\partial \mathcal{L}}{\partial x} = \frac{\partial \mathcal{L}}{\partial x_b}. \quad (14)$$

The experimental results of using different methods to deal with the sign function are shown in Tab. 3. The original STE method performs best among different methods. This conclusion is different from that in BNNs. We analyze that this is because both XNOR-Net (Rastegari et al., 2016) and Dorefa-Net (Zhou et al., 2016) modify the forward pass of sign function by adding per-channel scales or a constant scale, which contaminate the direction group and make it unable to focus on giving the correct binary outputs.

Table 3: Different methods are used to deal with sign function. The experiments are conducted on LLaMA-7B and the commonsense reasoning dataset.

| Method | Trainable Params (%) | Avg. |
|---|---|---|
| STE (ours) | 1.64 | 78.9 |
| XNOR-Net | 1.64 | 77.9 |
| Dorefa-Net | 1.64 | 78.1 |

**The influence of $r_1$ and $r_2$.** As shown in Sec. 3.3, $r_1$ and $r_2$ are the ranks of the low-rank matrices in the magnitude and direction groups, respectively. In previous experiments, we set $r_1 = r_2 = r$ in default to avoid introducing new hyper-parameters compared to other LoRA-based methods. Thus, in this section we dive deeper into the influence of $r_1$ and $r_2$.

Specifically, we keep the total trainable parameters unchanged by setting $r_1 + r_2 = 2r$, and adjust the ratio of parameters in the magnitude group and direction group, which are controlled by $r_1$ and $r_2$, respectively. We conduct experiments on the commonsense reasoning dataset with LLaMA3-8B as our baseline model and set 15 different ratios. Other training hyper-parameters are kept the same as in the previous experiments. The results are shown in Fig. 2. We can see that the proposed Dual LoRA can consistently outperform LoRA and DoRA when having a roughly balanced parameter ratio between the magnitude group and direction group (from 25% to 70%), which shows the robustness of our method.

More ablation studies are shown in Appendix B.

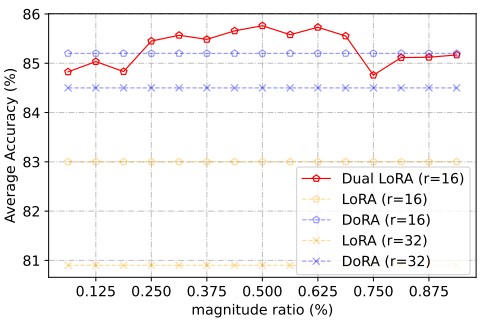 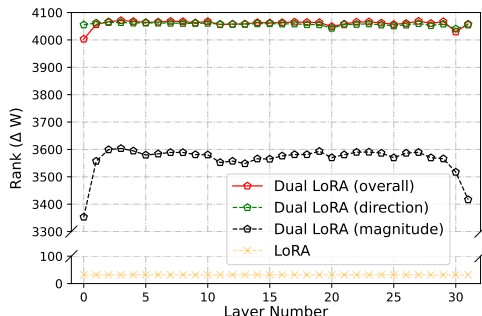

Figure 2: Average accuracy on the common-sense reasoning datasets using LLaMA3-8B as the baseline model with $r_1 = \{2, 4, \cdots, 30\}$ and $r_2 = 32 - r_1$ in the experiments. The red line is the proposed Dual LoRA, the blue/orange lines represent DoRA/LoRA with different ranks.

Figure 3: The average rank of $\Delta W$ for LoRA, magnitude group of Dual LoRA, direction group of Dual LoRA, and the overall Dual LoRA. The experiments are conducted on LLaMA2-7B.

## 5 ANALYSIS OF THE RANK OF THE UPDATE MATRIX

In this section, we given an analyze of the rank of the update matrix Eq. 11 to further show the priority of our method. Note that for a given matrix $X \in \mathbb{R}^{m \times n}$, $\text{Rank}(x) \leq \min(m, n)$ always holds true. Thus, in the original LoRA, given $A \in \mathbb{R}^{r \times k}$ and $B \in \mathbb{R}^{d \times r}$ with $r \ll \min(d, k)$, the rank of the update matrix $\Delta W = \frac{\alpha}{r} \cdot BA$ is upper bounded by:

$$\text{Rank}(\Delta W) = \text{Rank}(BA) \leq min(\text{Rank}(A), \text{Rank}(B)) \leq r. \tag{15}$$

In the proposed Dual LoRA, however, we found that the rank of the update matrix can achieve a higher upper bound. Specifically, given two low-rank matrices $A \in \mathbb{R}^{r_1 \times k}$ and $B \in \mathbb{R}^{d \times r_1}$ in the magnitude group and two low-rank matrices $C \in \mathbb{R}^{r_2 \times k}$ and $D \in \mathbb{R}^{d \times r_2}$ in the direction group with $r_1, r_2 \ll \min(d, k)$, the rank of the update matrix $\Delta W' = \frac{\alpha}{\sqrt{r_1 r_2}} \text{ReLU}(BA) \odot \text{Sign}(DC)$ is:

$$\begin{aligned}
\text{Rank}(\Delta W') &= \text{Rank}(\text{ReLU}(BA) \odot \text{Sign}(DC)) \\
&\leq \text{Rank}(\text{ReLU}(BA)) \times \text{Rank}(\text{Sign}(DC)) \\
&\leq \min(k, d)^2. \tag{16}
\end{aligned}$$

Note that the $\text{ReLU}(\cdot)$ and $\text{Sign}(\cdot)$ operations break the low-rank limitation of the original input matrix and derive output matrices with high rank.

In Fig. 3, we explicitly show the average rank of the update matrix in LoRA and the proposed Dual LoRA (magnitude group, direction group, and overall) over different layers. The experiments are conducted on LLaMA2-7B. We can see that the update matrix and the direction group almost achieve full rank (4096). The magnitude group has a relatively lower rank but is still much larger than that in LoRA. The results show the priority of our method from the perspective of matrix rank.

## 6 CONCLUSION

Original LoRA and its followers fine-tune the model without incorporating any prior knowledge and share a common drawback: as the trainable parameters are limited, the model accuracy is unsatisfactory. In this paper, we propose a new LoRA-based method called Dual LoRA, which incorporates an inductive bias into the original LoRA and improve the performance by introducing four low-rank matrices and separating them into the magnitude group and the direction group. The former controls the amplitude and whether or not we should update a parameter, and the latter decides whether or not this parameter should be updated in a positive or negative direction. Parameters in two groups are combined together to simulate the parameter updating process of FFT with gradient-based optimization methods. Experimental results on a wide range of NLP tasks and baseline models show that our Dual LoRA can consistently outperform LoRA and other state-of-the-art methods such as LoRA+ and DoRA with the same or less number of trainable parameters.

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

# A Experiments on Neural Language Generation (NLG)

**Datasets.** We conduct experiments on E2E NLG Challenge (Novikova et al., 2017), DART (Nan et al., 2020) and WebNLG (Gardent et al., 2017) datasets. The E2E dataset consists of approximately 42,000 training data, 4,600 validation data, and 4,600 test data. Each sample is composed of a sequence of slot-value pairs $(x, y)$ and a corresponding natural language reference text. DART is an open-domain data-to-text dataset with around 82,000 samples, each sample is structured as a sequence of entity-relation-entity triple. WebNLG has a total of 22,000 examples from 14 different categories, each sample is structured as a sequence of subject-property-object triple.

Table 4: The results of the proposed Dual LoRA and other competitors with GPT2_M and GPT2_L on the E2E NLG Challenge dataset. For all matrices, higher is better.

| Model | Methods | Trainable Parameters (M) | E2E NLG Challenge | | | | |
| | | | BLEU | NIST | MET | ROUGE-L | CIDEr |
|---|---|---|---|---|---|---|---|
| GPT2_M | FFT | 354.92 | 68.2 | 8.62 | 46.2 | 71.0 | 2.47 |
| | Adapter-L | 11.09 | 68.9 | 8.71 | 46.1 | 71.3 | 2.47 |
| | Prefix | 0.35 | 69.7 | 8.81 | 46.1 | 71.4 | 2.49 |
| | LoRA ($r = 4$) | 0.35 | 68.9 | 8.69 | 46.4 | 71.4 | 2.52 |
| | LoRA ($r = 8$) | 0.70 | 69.9 | 8.77 | 46.8 | 71.7 | 2.50 |
| | LoRA+ ($r = 8$) | 0.70 | 70.2 | 8.81 | 46.6 | 71.6 | 2.53 |
| | DoRA ($r = 8$) | 0.71 | 69.5 | 8.75 | 46.4 | 71.4 | 2.52 |
| | Dual LoRA ($r = 4$) | 0.70 | **70.6** | **8.86** | **46.9** | **72.4** | **2.56** |
| GPT2_L | FFT | 774.03 | 68.5 | 8.78 | 46.0 | 69.9 | 2.45 |
| | Adapter-L | 23.00 | 69.1 | 8.68 | 46.3 | 71.4 | 2.49 |
| | Prefix | 0.77 | 70.3 | 8.85 | 46.2 | 71.7 | 2.47 |
| | LoRA ($r = 4$) | 0.77 | 70.3 | 8.85 | 46.8 | 71.9 | 2.52 |
| | LoRA ($r = 8$) | 1.54 | 70.0 | 8.80 | 46.8 | 71.7 | **2.54** |
| | LoRA+ ($r = 8$) | 1.54 | 70.0 | 8.83 | 46.8 | 71.9 | 2.53 |
| | DoRA ($r = 8$) | 1.56 | 69.8 | 8.78 | 46.6 | 71.6 | 2.52 |
| | Dual LoRA ($r = 4$) | 1.54 | **70.6** | **8.86** | **47.1** | **72.5** | **2.54** |

Table 5: The results of the proposed Dual LoRA and other competitors with GPT2_M and GPT2_L on DART and WebNLG datasets. The up arrow indicates that the higher is better, and the down arrow indicates that the lower is better.

| Model | Methods | Trainable Parameters (M) | DART | | | WebNLG | | |
| | | | BLEU↑ | MET↑ | TER↓ | BLEU↑ | MET↑ | TER↓ |
|---|---|---|---|---|---|---|---|---|
| GPT2_M | LoRA ($r = 4$) | 0.35 | 47.4 | **0.36** | **0.47** | 55.0 | 0.37 | 0.39 |
| | LoRA ($r = 8$) | 0.70 | 47.5 | **0.36** | **0.47** | 55.6 | **0.38** | 0.39 |
| | LoRA+ ($r = 8$) | 0.70 | 47.6 | **0.36** | **0.47** | 56.1 | **0.38** | 0.39 |
| | DoRA ($r = 8$) | 0.71 | 47.0 | **0.36** | 0.48 | 53.5 | 0.36 | 0.40 |
| | Dual LoRA ($r = 4$) | 0.70 | **48.3** | 0.36 | **0.47** | **56.6** | **0.38** | **0.38** |
| GPT2_L | LoRA ($r = 4$) | 0.35 | 47.7 | **0.36** | **0.47** | 57.1 | 0.37 | **0.38** |
| | LoRA ($r = 8$) | 0.70 | 47.5 | **0.36** | **0.47** | 57.6 | 0.38 | **0.38** |
| | LoRA+ ($r = 8$) | 0.70 | 47.7 | **0.36** | **0.47** | 57.5 | 0.38 | **0.38** |
| | DoRA ($r = 8$) | 0.71 | 47.2 | **0.36** | **0.47** | 57.4 | **0.39** | **0.38** |
| | Dual LoRA ($r = 4$) | 0.70 | **48.4** | 0.36 | **0.47** | **57.7** | **0.39** | **0.38** |

**Results.** As in previous experiments, we reduce our rank to half of the other LoRA-based methods to ensure the same number of trainable parameters. All other hyper-parameters are the same as LoRA, except that we tune the learning rate. As the results shown in Tab. 4 and Tab. 5, Dual LoRA can consistently outperform all other competitors on three different datasets with baseline model GPT2_M and GPT2_L.

# B MORE ABLATION STUDIES

In this section, we conduct more ablation studies on the proposed Dual LoRA.

## B.1 DIFFERENT FORMS OF THE UPDATE MATRIX

Specifically, we investigate the following settings:

- Setting 1: Remove $\text{ReLU}(\cdot)$ function in Dual LoRA, which means $\Delta W = \frac{\alpha}{\sqrt{r_1 r_2}}(BA) \odot \text{Sign}(DC)$.
- Setting 2: Remove $\text{Sign}(\cdot)$ function in Dual LoRA, which means $\Delta W = \frac{\alpha}{\sqrt{r_1 r_2}}\text{ReLU}(BA) \odot (DC)$.
- Setting 3: Replace the output of the direction group with a random-initialized binary matrix and fix this matrix during training. Given $W_b$ as the random initialized binary matrix, we have $\Delta W = \frac{\alpha}{r_1}\text{ReLU}(BA) \odot W_b$.

Table 6: The results of the previous settings on the commonsense reasoning dataset, with LLaMA-7B as the base model.

| Methods | Trainable Param. (%) | Commonsense Reasoning Datasets | | | | | | | | |
|---|---|---|---|---|---|---|---|---|---|---|
| | | BoolQ | PIQA | SIQA | HellaS | WinoG | ARC-e | ARC-c | OBQA | Avg. |
| Dual LoRA ($r=32$) | 1.64 | 70.0 | **83.2** | **79.5** | **87.3** | 83.0 | **81.6** | 65.2 | **81.0** | **78.9** |
| Setting 1 ($r=32$) | 1.64 | 70.3 | 82.9 | 78.8 | 85.4 | 83.3 | 82.1 | **65.2** | 79.4 | 78.4 |
| Setting 2 ($r=32$) | 1.64 | **70.6** | 80.8 | 78.9 | 84.7 | 81.8 | 80.4 | **65.2** | 78.8 | 77.7 |
| Setting 3 ($r=32$) | 0.84 | 60.4 | 0.2 | 1.7 | 0.2 | 0.2 | 0.5 | 0 | 0 | 7.9 |
| Setting 3 ($r=64$) | 1.64 | 36.3 | 36.2 | 6.6 | 0 | 14.3 | 4.9 | 11.5 | 0 | 13.7 |

We conduct experiments on LLaMA-7B and perform an evaluation on the commonsense reasoning dataset. The results are shown in Tab. 6. We can see that removing $\text{ReLU}(\cdot)$ and $\text{Sign}(\cdot)$ functions cause marginal performance drop, and using a random initialized binary matrix to replace the direction group severely degrades the performance, which shows the importance of the proposed architecture.

## B.2 DIFFERENT ACTIVATION FUNCTIONS FOR THE MAGNITUDE GROUP

In the main paper, we use $\text{ReLU}(\cdot)$ function for the magnitude group. There are other functions that can keep the activation greater than zero, such as $\text{Abs}(x) = |x|$ and $\text{Sigmoid}(x) = 1/(1 + e^{-x})$. We compare them in Tab. 7 on the commonsense reasoning dataset with LLaMA-7B model.

Table 7: The results of different activation functions for the magnitude group on the commonsense reasoning dataset, with LLaMA-7B as the base model.

| Activation | Commonsense Reasoning Datasets | | | | | | | | |
|---|---|---|---|---|---|---|---|---|---|
| | BoolQ | PIQA | SIQA | HellaS | WinoG | ARC-e | ARC-c | OBQA | Avg. |
| ReLU (ours) | **70.0** | **83.2** | **79.5** | **87.3** | **83.0** | **81.6** | 65.2 | **81.0** | **78.9** |
| Abs | 68.6 | 82.9 | 77.9 | 74.3 | 74.0 | **81.6** | **65.8** | 79.6 | 75.6 |
| Sigmoid | 66.5 | 80.3 | 78.5 | 77.8 | 79.4 | 75.4 | 60.9 | 76.6 | 74.4 |

We can see that the original setting with $\text{ReLU}(\cdot)$ function performs the best. This is because neither the $\text{Abs}(\cdot)$ function nor the $\text{Sigmoid}(\cdot)$ function can flexibly give zero output. Note that zero output means that we can easily freeze some of the elements that are already well-trained for the downstream tasks in the original weight matrix, which is the advantage of the $\text{ReLU}(\cdot)$ function.

