# OpenReview forum: "Dual LoRA: Enhancing LoRA with Magnitude and Direction Updates"
_ICLR.cc/2026/Conference — ICLR 2026 Conference Withdrawn Submission_

### Official Review · Reviewer_ADJx · 2025-10-24

**Soundness:** 3
**Presentation:** 4
**Contribution:** 3
**Rating:** 4
**Confidence:** 5

**Summary:**

This work proposes Dual LoRA, which splits LoRA's low-rank matrices into magnitude (ReLU) and direction (Sign+STE) groups to simulate full fine-tuning. It outperforms LoRA and its SOTA variants. Experiments on NLG/NLU/commonsense reasoning with GPT-2, RoBERTa, LLaMA-1/2/3 validate its effectiveness with the same trainable parameters.

**Strengths:**

1.The paper is well organized and well written.

2.The authors present a well-motivated approach with a simple, easy-to-follow framework.

3.It conducts numerous experiments, validates the experimental results on models of various series and sizes, and covers a wide range of evaluation tasks

**Weaknesses:**

1. **In Section 5, the analysis of rank is not rigorous.**
   The operations of ReLU and Sign can both potentially reduce the rank, so the inequality in Equation (16) only holds in certain cases. The authors should either limit its validity scope or clarify, perhaps with reference to Figure 3, that the inequality holds in most cases but not universally. Furthermore, using Equation (16) to claim an upper bound on the rank is not meaningful, since Equation (15) similarly satisfies $rank(BA)\le rank(B)\times rank(A)$ (when the ranks of (A) and (B) are non-zero), and the upper bound scaled by $min(k,d)^2$ is too loose.

2. **The paper lacks detailed reporting of experimental settings.**
   Important hyperparameters such as learning rate, number of epochs, and the coefficient $alpha$ are missing. It is also unclear which modules LoRA is applied to (e.g., query, key), and how checkpoints are selected. Even if the authors follow prior settings, reporting these details is essential for reproducibility and for future work to follow up.

3. **The paper does not explain the performance gap between PEFT and FFT methods.**
   In Tables 2 and 4, PEFT-based methods outperform FFT, and even LoRA with (r=4) achieves better results than FFT in Table 4. The paper lacks any discussion of this phenomenon. Considering the insufficient description of experimental details, this discrepancy may result from incomplete convergence, and the authors should discuss the convergence behavior of the compared methods.

4. **Appendix B.1 lacks an ablation study removing both ReLU and Sign.**
   Such an experiment is necessary to fully isolate the effect of these nonlinearities on the model’s rank and performance.

**Questions:**

1.Since this method effectively enlarges the rank space during fine-tuning, it may show advantages when the rank is relatively low. However, as the rank increases (e.g., (r=32, 64, 128, 256,) etc.), will this advantage diminish?

---

### Official Review · Reviewer_GAJ4 · 2025-10-26

**Soundness:** 1
**Presentation:** 4
**Contribution:** 1
**Rating:** 2
**Confidence:** 4

**Summary:**

In this paper, the authors try to address LoRA's limitation: its low-rank assumption restricts the SFTed models' performance. To achieve this goal, the authors proposed Dual LoRA, a LoRA variant that enhances LoRA by simulating the gradient-based parameter updates of FFT. Specifically, Dual LoRA separates LoRA's low-rank matrices into two functionally distinct groups, magnitude and direction, each optimized to control a specific aspect of weight updates.

**Strengths:**

1. The paper is well-written and easy to read.

**Weaknesses:**

1. To address the disadvantage of LoRA and its variants, the authors separated the low-rank matrices into two groups: the magnitude and direction groups. While in the experiments, the authors did not validate the effectiveness of the magnitude group.

2. To address the problem brought by fewer training parameters of LoRA, the authors introduce two groups of low-rank matrices. The proposed method obviously increases the computational cost compared to LoRA, while the authors did not provide the computational cost analysis.

3. The authors did not provide evidence that the improved performance does not come from the additional low-rank matrices.

**Questions:**

Please see the weaknesses.

---

### Official Review · Reviewer_B3qP · 2025-10-28

**Soundness:** 2
**Presentation:** 2
**Contribution:** 2
**Rating:** 2
**Confidence:** 4

**Summary:**

In this paper, the authors argue that standard LoRA may be sub-optimal because it lacks an explicit inductive bias about how parameters should be updated. To address this, they propose separating LoRA updates into magnitude (via ReLU) and direction (via Sign) components. They replace the standard matrices B and A in LoRA with four low-rank matrices, paired with the nonlinear operations ReLU and Sign, combined via element-wise multiplication. Experiments across commonsense reasoning and NLU tasks show that Dual LoRA consistently outperforms both standard LoRA and DoRA.

**Strengths:**

- The proposed Dual LoRA uses two groups of parameters to incorporate an inductive bias, which seems interesting.
- They evaluate Dual LoRA from 7B up to 70B model sizes and consistently outperform LoRA and DoRA.
- They introduce warmup tricks and straight-through estimator (STE) tricks.

**Weaknesses:**

- This idea seems closely mirror DoRA (ICML 2024), which also decomposes weight updates into magnitude and direction. Although there are slight differences in how to model and combine the magnitude and directions (four low-rank matrices and element-wise multiplication in this paper, versus two matrix multiplications in the DoRA paper), the distinction does not fundamentally alter the underlying paradigm.

- The evaluation is limited.
  - They only compare with DoRA/LoRA on generation tasks and include uncommon baselines like GaLore, RandLoRA, and VeRA. VeRA trains only small vectors to extremely reduce rank, so it is expected to perform worse, the comparison is therefore unfair. Advanced methods like LoRAPro, LoRA-GA, and PiSSA are missing.
  - Furthermore, for generation tasks, they evaluate only on short and easy QA tasks , which cannot fully reflect the method’s performance. In fact, the original DoRA evaluates on the common 170k dataset to ensure robustness.
  - They also exclusively use the LLaMA series, which is outdated; newer models like Qwen should be discussed.
  - The paper states: “We tune the learning rate for our method, and all other training hyper-parameters are kept unchanged as in DoRA”.  This does not seem fair because the experimental settings are clearly different from DoRA (e.g., different datasets, different ranks).

- Equation 16 is too loose. ΔW is in $\mathrm{R}^{k\times d}$, so its rank must be smaller than $min(k, d)$, claiming a bound of $min(k, d)^2$ is mathematically meaningless and misleading.

- They use $ReLU(BA)$ to implement "freeze well-trained elements." However, in the original LoRA, after computing BA in the FFN, they also apply $ReLU(W + BA)$, which could similarly suppress updates to well-performing directions. Moreover, directly wrapping $BA$ with ReLU may inadvertently zero out useful low-magnitude updates, potentially harming optimization.

- The authors claim “with fewer trainable parameters,” yet they do not mention that the two low-rank multiplications ($BA$ and $DC$) combined via element-wise Hadamard product ($BA \odot DC$) are likely much costlier. The paper provides no training-time, memory, or throughput metrics and does not analyze merge or inference overhead.



- Nonlinear element-wise functions (ReLU, Sign) break linear rank properties. For example, if BA is a random matrix, applying ReLU zeros out roughly half of its entries, potentially destroying its rank. With the sign operation, convergence properties cannot remain stable or favorable. In fact, the authors also observe that without warmup tricks, training can collapse.

- Given the above three points and the marginal improvement (e.g., 91.9 vs. 91.7 for MNLI in Table 2), I believe the performance gain does not justify the method’s complexity. Their assertion of achieving “state-of-the-art under the same or fewer parameters” is unsupported and overclaimed.

**Questions:**

see Weaknesses

---

### Official Review · Reviewer_msyo · 2025-10-29

**Soundness:** 2
**Presentation:** 3
**Contribution:** 2
**Rating:** 2
**Confidence:** 2

**Summary:**

The paper attempts to improve LoRA by replacing the $\Delta W = BA$ with in LoRA with a formulation of direction times magnitude, i.e. $\Delta W = ReLU(BA) \cdot Sign(DC)$, which allows higher rank $\Delta W$ and leads to empirically improved results.

**Strengths:**

The formulation appears to be novel and empirically useful. The presentation is clear and easy to follow.

**Weaknesses:**

The motivation behind this formulation is quite unclear. The paper mentioned that the actual update can be represented as direction * magnitude, but we can represent $\Delta W$ in arbitrary way and I can't see why the proposed way is preferred. On the other hand, the whole Sign(DC) part only produces 1-bit information, which appears to be quite restrictive. My own feeling is that the proposed method appears to be a special case of a gated version of LoRA similar to GLU, with constraints of non-negative values and $\pm$1 gate. It is not surprising that adding gates with non-linearity leads to a higher rank upper bound and better performance, but a more general gate may enable even better results; currently the empirical improvement is not so large.

**Questions:**

L218-220: Is it really a special advantage of the proposed method? Standard LoRA allows zero updates as well, and this is how it is initialized. More empirical evidence is necessary to support this claim.

L249-252: Then the question becomes will there be a lot of dead/zero items in $\Delta W$? I'm quite curious about the actual distribution of ReLU(BA) and Sign(DC) in different tasks and layers.

L269: What is the warm-up strategy?

With more constraints dual LoRA with rank=r should have smaller representational capacity than LoRA with 2r, and we know that LoRA leads to better performance on small datasets cf. full fine-tuning because it serves as regularization with smaller capacity. Will the advantage simply because it happens to better match the required capacity of certain tasks than a r-ranked LoRA/DoRA or a 2r-ranked one? This is particularly possible in experiments where increasing r doesn't lead to better results.

How much training costs does it add compared to LoRA, DoRA, etc.? Maybe the direction group can be calculated with lower precision.

L453: given -> give/gave
L454: priority -> superiority?

---

### Official Review · Reviewer_zQB7 · 2025-10-31

**Soundness:** 2
**Presentation:** 2
**Contribution:** 2
**Rating:** 4
**Confidence:** 4

**Summary:**

This paper proposes Dual LoRA, a new parameter-efficient fine-tuning (PEFT) method for large language models (LLMs). Dual LoRA uses four low-rank matrices in total: two forming a “magnitude group” processed by a ReLU, and two forming a “direction group” processed by a sign function (with straight-through estimator). This decomposition is designed to simulate the gradient-based parameter update behavior of full fine-tuning (FFT). Comprehensive experiments on NLG, NLU, and commonsense reasoning tasks (e.g., GLUE, BoolQ, HellaSwag, DART) across GPT-2, RoBERTa, DeBERTa, and LLaMA 1/2/3 (up to 70B) demonstrate consistent improvements over LoRA, LoRA+, DoRA, and other SOTA PEFT methods with the same number of trainable parameters.

**Strengths:**

1. The idea of splitting LoRA updates into magnitude and direction components is simple. It introduces an inductive bias aligned with how FFT updates parameters.
2. Dual LoRA only adds two extra low-rank matrices and simple nonlinearities (ReLU, Sign).
3. The paper includes extensive experiments on diverse tasks and model scales (7B–70B), showing consistent and sometimes notable gains. And the ablation study on STE variants (XNOR-Net, DoReFa-Net) and rank ratios (r₁/r₂) is detailed and insightful.

**Weaknesses:**

1. Although the paper claims Dual LoRA mimics FFT’s parameter update process, it does not quantify this resemblance. Adding empirical analysis comparing Dual LoRA’s update direction to FFT will enhance this paper.
2. Both Dual LoRA and DoRA employ magnitude–direction decomposition. While Section 2.2 discusses differences, the distinction remains conceptually blurry. It is necessary to include a structural comparison figure or ablation (e.g., removing ReLU/Sign) to explicitly show what Dual LoRA contributes beyond DoRA.
3. The paper lacks theoretical justification for why separating a low-rank structure into two subspaces (magnitude and direction) is beneficial. A more rigorous theoretical or mathematical analysis explaining this design choice would enhance the overall contribution.
4. The performance improvement is relatively modest. For instance, on LLaMA3-8B, the gain is only +0.6% compared with DoRA, which could fall within the range of random variance. It is therefore important to report standard deviations or conduct statistical significance tests across multiple runs to ensure the robustness of the results.

**Questions:**

Please refer weaknesses.

---

### Note · Authors · 2026-01-05

I have read and agree with the venue's withdrawal policy on behalf of myself and my co-authors.